# Evaluation of Residents’ Timing of Return to or New Settlement in Kawauchi Village, at 10 Years after the Fukushima Daiichi Nuclear Power Plant Accident

**DOI:** 10.3390/ijerph19010543

**Published:** 2022-01-04

**Authors:** Hitomi Matsunaga, Makiko Orita, Mengjie Liu, Yuya Kashiwazaki, Yasuyuki Taira, Noboru Takamura

**Affiliations:** Department of Global Health, Medicine and Welfare, Atomic Bomb Disease Institute, Nagasaki University, Nagasaki 8528523, Japan; orita@nagasaki-u.ac.jp (M.O.); liumengjie19940221@163.com (M.L.); y-kashiwazaki@nagasaki-u.ac.jp (Y.K.); y-taira@nagasaki-u.ac.jp (Y.T.); takamura@nagasaki-u.ac.jp (N.T.)

**Keywords:** Fukushima Daiichi nuclear accident, life satisfaction, SOC-13

## Abstract

Kawauchi village in Fukushima prefecture was affected by the Fukushima Daiichi Nuclear Power Plant (FDNPP) accident, and residents had to evacuate from their hometown in 2011. This study clarified the timing and related factors with regard to residents returning to or newly settling in Kawauchi. A survey was conducted using a questionnaire, from February to March 2021, with assistance from the Kawauchi village office and post office. Of the 374 residents, 170 (45.5%) had returned to or newly settled in Kawauchi within the past 2 years (group 1), 84 (22.5%) in the past 2–5 years (group 2), and 99 (26.5%) after more than 5 years (group 3) following the evacuation order. An additional 21 residents (5.5%) who had lived in Kawauchi at the time of the FDNPP had not yet returned (group 4). Compared with the other groups, residents in group 1 were more satisfied with their current lives and were coping better with stress. Even though they had experienced a serious nuclear disaster, residents of Kawauchi village who returned to their hometown in the early phase had a high sense of satisfaction with their current life one decade after the FDNPP accident.

## 1. Introduction

In March 2011, the Tokyo Electric Power Company’s Fukushima Daiichi Nuclear Power Plant (FDNPP) accident occurred as a result of the Great East Japan Earthquake and the massive tsunami of over 40 m, which is the largest recorded since observations began (magnitude 9.0). This complex disaster caused extreme direct damage occurred due to destruction of public equipment, as well as disruption of life and business activities, interruption of lifelines, and severance of supply chains [1,2]. More serious was the release of various radionuclides exceeding standard values into the environment from the FDNPP. Corresponding to the nuclear accident, the government ordered residents in a 3 km radius around the FDNPP to evacuate on 11 March 2011. As the seriousness of the accident became apparent, the areas of evacuation orders were gradually expanded. On 22 April 2011, the government of Japan decided to rearrange the areas to which evacuation orders have been issued into 3 areas, depending on the annual integrated doses and the distance from FDNPP, as follows: evacuation order zone (warning zone), planning evacuation zone, and, emergency evacuation preparation zone (evacuation instruction canceled on 30 September 2011). These consequences necessitated the long-term evacuation of approximately 164,825 people from their hometown: 102,827 people in the area of Fukushima and 62,038 people outside Fukushima [3]. Moreover, evacuees consequently suffered chronic mental and physical health due to the accident, manifesting as depression and increased levels of cardiovascular and lifestyle-related diseases [4,5,6].

In March 2012, the government rearranged only depending on the annual integration, as follows: difficult to return zone (annual integrated doses are over 50 mSv), restricted residence zone (annual integrated doses are between 20 to 50 mSv), and evacuation order cancellation preparation zone (annual integrated doses are below 20 mSv). In addition, then, areas of below 20 mSv gradually lifted evacuation orders after considering the period for preparing tasks, such as confirmation of infrastructure and security. However, the longer the duration of evacuation, the greater the hesitation of people to return from their evacuation site to the area affected by the nuclear accident. Reasons for low rates of intention to return included employment mismatch for themselves, education for their children, and anxiety regarding the health effects of radiation exposure [7]. We have reported previously that the residents of Tomioka, which is located within 20 km of the FDNPP, had started to return in 2016. Approximately 65% of former residents responded that they had decided not to return to Tomioka. One of the reasons for their decision was concern about the effect of radiation on the health of their genetic effects in offspring [8,9]. As of December of 2021, the residence rate of Tomioka (number of people who had returned to or settled in Tomioka/number of people who had resident registration) was approximately 15% (1803/12,066 residents) [10] due to complex personal or social circumstances of the former residents, for example, having a stable livelihood at the evacuation site or due to health conditions [11].

Kawauchi village is located within approximately 30 km of the FDNPP and had a population of approximately 3000 at the time of the accident (Figure 1). Immediately after the FDNPP accident, during the confusion following damage to the village from the severe earthquake, Kawauchi village accepted evacuees from Tomioka. However, on 14 March 2011, the government of Japan issued an order for all people within a 30 km radius of the FDNPP to stay indoors, and then the Mayor of Kawauchi issued an evacuation order to Koriyama city, Fukushima, which is about 60 km away from Kawauchi, for all residents on 16 March 2011. Approximately 6 months after the evacuation order was issued, the only area of the emergency evacuation preparation zone was lifted on 30 September 2011. In addition, then, approximately 9 months after the evacuation order was issued, the Mayor of Kawauchi decided to return to the Kawauchi areas more than 20 km from the FDNPP (the area where evacuation order was lifted as an emergency evacuation preparation zone), which included the main residential area. After that, the evacuation order was gradually lifted for areas of Kawauchi less than 20 km from the FDNPP, with the evacuation order for all areas of Kawauchi lifted in June 2016, approximately 5 years after the FDNPP accident following the results of the assessment of radioactive contamination [12]. The result of such an incredibly early lifting of the evacuation order was that the number of residents of Kawauchi gradually increased. By 2018, residents of Kawauchi numbered approximately 80% of the pre-disaster total [13,14]. This study aimed to clarify the timing of subjects’ return to, or new settlement in, Kawauchi village and related factors at just 10 years after the FDNPP accident. The return rate was extremely high in Kawauchi village, even with a small number of the residents compared to the area where evacuation was lifted from the FDNPP accident. The results contribute to the evaluation that the early evacuation order was lifted from a nuclear accident and benefitted the affected area of residents, whether they returned or not, and the stakeholders who comprehensively support them.

## 2. Methods

### 2.1. Study Participants

The present survey was conducted in February and March 2021, just 10 years after the FDNPP accident in Kawauchi village, Fukushima prefecture. The total population of Kawauchi in December of 2021 was 2439 people (1237 males and 1202 females), of whom 431 were evacuees (229 males and 202 females) [15]. Details of the study sets have been described previously [16]. Briefly, we defined residents of Kawauchi as those who held a resident card, even if they had not yet returned Kawauchi, as well as those who were living in Kawauchi without a resident card, at the time of the survey. We distributed the study questionnaire to Kawauchi residents aged >20 years via mail with the cooperation of the Kawauchi public office and the Kawauchi post office. Of the 451 total responses obtained, 374 (204 male and 170 female) were regarded as valid after excluding incomplete responses. Prior to the study, we obtained permission from the municipal government of Kawauchi to implement this study. We explained the study using a leaflet and obtained informed consent from all participants. This study was approved by the ethics committee of the Nagasaki University Graduate School of Biomedical Sciences (No. 20112703).

### 2.2. Data Collection

The questionnaire was developed based on those employed in our previous studies conducted on areas of Fukushima Prefecture which had their evacuation order lifted [8,17]. Those studies surveyed risk perception regarding radiation exposure and its health effects, and a mental health and lifestyle survey was conducted within the framework of the Fukushima Health Management Survey, which was organized by Fukushima Prefecture [6,18]. In the questionnaire of new components of the study was the timing of return or new settlement, estimate their sense of value in Kawauchi, and life satisfaction with their current life. 

To assess the characteristics of the residents according to differences in the timing of return or new settlement in Kawauchi village, we defined group 1 as “those who had returned or started to live in Kawauchi within 2 years of the evacuation order (March 2011 to March 2013)”, group 2 as “those who had returned or started to live in Kawauchi at 2–5 years after the evacuation order (April 2013 to March 2016)”, group 3 as “those who had returned or started to live in Kawauchi more than 5 years after the evacuation order (after March 2016)”, and group 4 as “not yet returned”. We collected the following demographic data: sex, age, living, and birthplace of residence in Kawauchi or elsewhere, place of residence at the time of the FDNPP accident (Kawauchi or elsewhere), and cohabitation with children of any age (or not). To estimate their sense of value in Kawauchi, we asked, “Do you think that Kawauchi village has been adequately reconstructed after the FDNPP accident?” and “Do you feel that you belong to the Kawauchi village community?” (“yes” and “probably yes” were scored as Yes, and “probably not” and “no” were scored as No). A question about whether the residents were concerned about consuming locally sourced foods in Kawauchi was also included. Furthermore, we assessed residents’ perception of the potential health risks of radiation from the FDNPP accident, such as the risk of cancer and genetic effects on the next generation.

To assess mental health, we included questions regarding life satisfaction from the post-traumatic stress disorder (PTSD) checklist (PCL-S) and sense of coherence (SOC)-13 checklist. Life satisfaction was evaluated by the question “Are you satisfied with your current life?” (‘satisfied’ and ‘slightly satisfied’ were scored as “satisfied”; ‘slightly unsatisfied’ and ‘unsatisfied’ were scored as “unsatisfied”). The PCL-S is widely used to assess the severity of traumatic reactions and to screen for PTSD [19,20]. The psychometric and screening properties of the PCL-S have been reported previously [21]. We used the standard cutoff of ≥12 as an indicator of mood/anxiety disorders, as defined in previous Japanese studies. The SOC-13 is used to assess capacity to cope with stress and has been widely used in epidemiological and psychological investigations [22]. It comprises 13 items with answers presented in a seven-point Likert format and is scored as the sum of all items (possible score, from 13 to 91). The higher the score, the stronger the SOC and ability to cope with stress. In this study, SOC was dichotomized as ≤59 and >59, which is the Japanese standard value [22,23].

### 2.3. Statistical Analyses

After the descriptive statistics of the residents’ characteristics were calculated, the chi-square test was used to investigate the difference in factors regarding the timing of return to or settlement in Kawauchi in each of the four groups. Regression analysis is one of the statistical analyses that can estimate the relationships between a dependent variable and various other variables. Then, regression analysis was conducted with group 3 as the dependent variable (people who had returned to or settled >5 years after the evacuation order) because group 3 returned to or settled after the evacuation order was lifted in all areas of Kawauchi. To take the regression analysis, we could compare the factors affecting timing of return or settlement between before and after the evacuation order had been lifted in all areas. *p*-values less than 0.05 were considered significant. Statistical analysis was performed using IBM SPSS Statistics Version 25 software (SPSS Japan, Tokyo, Japan) and SAS 8.2 software (SAS Institute, Cary, NC, USA).

## 3. Results

Table 1 lists the sociodemographic factors and results of the chi-square test for all groups. Of the 374 residents studied, the number who had returned or started to live in Kawauchi was 170 (45.5%) within 2 years (group 1), 84 (22.5%) at 2–5 years (group 2), and 99 (26.5%) at over 5 years (group 3), after the evacuation order. In addition, 21 (5.5%) of the residents who lived in Kawauchi at the time of the FDNPP had not yet returned (group 4). Residents in group 3 were significantly younger than those in groups 1, 2, and 4. Among the groups, there were fewer group 3 residents living in Kawauchi at the time of the FDNPP accident compared with those of other groups. In contrast, fewer residents in group 4 were born in Kawauchi compared with any other group. Residents’ sense of satisfaction with their current lives and their sense of belonging in Kawauchi was significantly higher in group 1 than in other groups, and the feeling that genetic effects would occur due to the FDNPP accident was significantly lower in group 1 than in other groups. The SOC-13 score was also significantly lower in group 3 than in other groups. There was no difference among the groups in terms of sex, cohabitation with children, opinion that Kawauchi has been adequately reconstructed after the FDNPP accident, concern about consuming wild plants growing in Kawauchi, the feeling that health effects would occur due to radiation exposure from the FDNPP accident, or frequency of PCL-S score ≥ 12.

Table 2 lists the results of regression analysis with group 3 as the reference. There was an independent association of age <60 years and place of birth being Kawauchi in group 1 (age, OR = 3.9, 95%CI: 2.1−7.0, *p* < 0.01; birthplace, OR = 2.2, 95%CI: 1.3−3.7, *p* < 0.01) and group 2 (age; OR = 4.5, 95%CI: 2.1−9.9, *p* < 0.01; birthplace; OR = 2.9, 95%CI: 1.5−5.6, *p* < 0.01) in comparison with group 3. Compared with group 3, a significantly higher percentage of residents in all other groups lived in Kawauchi at the time of the FDNPP accident: group 1, OR = 16.2, 95%CI: 6.9–38.0, *p* < 0.01; group 2, OR = 4.2, 95%CI: 2.0–8.7, *p* < 0.01; and group 4, OR = 3.8, 95%CI: 1.0–13.7, *p* < 0.05. Accordingly, there were more new residents after the FDNPP in group 3 than in other groups. Furthermore, there was an independent association of cohabitation with children in group 4 (OR = 3.1, 95%CI: 1.1−8.5, *p* < 0.05) in comparison with group 3. There was also an independent association of a feeling of satisfaction with their current lives and sense of belonging in Kawauchi in group 1 (satisfaction, OR = 4.0, 95%CI: 2.3–6.7, *p* < 0.01; a sense of belonging, OR = 2.3, 95%CI: 1.3–4.1, *p* < 0.01), group 2 (satisfaction, OR = 2.6, 95%CI: 1.4–4.8, *p* < 0.01; a sense of belonging, OR = 2.0, 95%CI: 1.0–3.8, *p* < 0.05) and group 4 (satisfaction, OR = 3.1, 95%CI: 1.1.–8.7, *p* < 0.05; a sense of belonging, OR = 0.3, 95%CI: 0.1–0.7, *p* < 0.01) compared with group 3. In contrast, there was an independent association of opinion that Kawauchi had been adequately reconstructed only in group 1 (OR = 1.8, 95%CI: 1.1−3.0, *p* < 0.01) in comparison with group 3. Risk perception of genetic effects in offspring caused by the FDNPP accident was significantly lower only in group 1 (OR = 0.4, 95%CI: 0.3–0.8, *p* < 0.01) compared with group 3. SOC-13 score was significantly higher in group 1 (OR = 2.7, 95%CI: 1.6–4.5, *p* < 0.01) and group 2 (OR = 2.2, 95%CI: 1.2–4.0, *p* < 0.01) compared with group 3. There was no significant difference among the groups in terms of sex, concern about consuming wild plants growing in Kawauchi, concern about health effects due to radiation exposure, or PCL-S score, compared with group 3.

## 4. Discussion

There were many difficulties to be overcome following the lifting of the evacuation order that was placed after the massive nuclear accident, including the need for decontamination, infrastructure development, and risk communication regarding radionuclides and their health effects on residents in the affected area [24]. Furthermore, the longer the time before the evacuation order was lifted, the more difficult it was for residents to return to the affected areas, as most evacuees started a new life elsewhere after they were evacuated from these areas [25]. In Kawauchi village, the residents started returning one year after the accident, depending on the degree of environmental radioactive contamination and the progress of decontamination work [12]. Therefore, the present study has investigated a unique and unprecedented situation following the lifting of an evacuation order at such an early stage after a serious nuclear accident. 

In past experiences of nuclear accidents, psychological and social consequences due to people’s anxiety about radiation exposure were the most serious problems, rather than physical health problems caused by direct radiation exposure. The social consequences included not only a broad socio-economic impact, such as an international reputation for fresh foods, etc., but also prejudice to the individual level of radiation health effects from the nuclear accident. Furthermore, previous studies reported that increased the various psychological impacts, such as the prevalence of hypertension, diabetes, dyslipidemia, and liver function, among the evacuees [26]. Especially, repeated evacuation and relocation following nuclear accidents have caused these serious psychological impacts and social consequences, particularly among vulnerable people, such as people who need some kind of care and elderly people [27,28]. Approximately one-third of the residents of Kawauchi villages was ≥65 years old at the time of the accident in 2011 [29]. Therefore, we thought that the risk of psychological and social consequences in the village was relatively high among residents of Kawauchi who had been evacuated and relocated after the nuclear accident. However, the present results revealed that group 1 had a higher level of satisfaction with their current lives and were coping better with stress compared with other groups, even though half of group 1 was aged >60 years at the time of the accident. Furthermore, compared with the other groups, group 1 had a higher sense of belonging in Kawauchi, and a higher proportion thought that Kawauchi had been reconstructed adequately, following the FDNPP accident. Therefore, the present results suggest that the early lifting of the evacuation order and return to their hometown had positive impacts for the evacuees, even though they lived in an area that had experienced a serious nuclear accident. In 2011, about a year after the FDNPP accident, external radiation exposure to residents was estimated to be sufficiently low based on external effective doses from soil samples [30]. Nagasaki University has started risk communication to residents who have returned to Kawauchi, based on such data as the evaluated external and internal radiation exposure doses [31,32]. Support for the evaluation of radiation exposure must be provided to residents in the initial phase, and radiation-dose-based risk assessment and communication are crucially essential.

It is noteworthy that the residents of group 1 had the lowest risk perception of genetic effects compared with the other groups. A previous study has revealed that those who were highly anxious about genetic effects due to the radiation exposure had low intention to return to their hometown after the FDNPP accident [8,33]. Although it is difficult to find a causal relationship between risk perception of genetic effects and the number of people who returned in the initial phase, we should have the attention of the result. 

We found that relative to other groups, residents in group 3 who had returned to or started living in Kawauchi more than 5 years after the lifting of the evacuation order were the youngest among the groups, and the group also had the lowest proportion of residents who were living in Kawauchi at the time of the FDNPP accident. These results showed that about half of the residents in group 3 had moved to and settled in Kawauchi after the FDNPP accident. Furthermore, the study results revealed that residents in group 3 had the lowest stress coping (SOC-13) score and the lowest satisfaction with current life compared with those in other groups. Kawauchi village has been active in promoting migration to the village and settlement of younger people and new residents after the nuclear accident [34]. Therefore, some of the group 3 residents may have moved there to take up job offers in a new industry following the accident. A previous study has reported a reduced capacity to cope with stress in the younger generation compared with the older generation [23]. Therefore, it is likely that the low SOC-13 score in group 3 was affected by the stress associated with each individual’s life stage, rather than by that due to the nuclear accident.

Compared with other groups, fewer residents of group 4 (who had not yet returned to the village) had been born in Kawauchi, although the percentage in this group who were living in Kawauchi at the time of the FDNPP was the same as in groups 1 and 2. Furthermore, the residents of group 4 had a low sense of belonging in Kawauchi but greater satisfaction in their current lives, even if those who have not returned. However, this result requires careful interpretation because of response bias, as only a small number of responses were received from residents in group 4. A previous study revealed that those who have not returned had higher stress than returnees [35]. In addition, there is a higher risk of mental health problems in people who have few emotional ties in mutually dependent social networks among vulnerable groups, such as the sick, the elderly, and children [36]. In contrast, the present result that some of the residents who chose not to return to Kawauchi were satisfied with their current lives suggests that their decisions not to return should be accepted and understood.

The present results found no significant difference between the timing of residents’ return to Kawauchi and their opinion regarding the safety of consuming wild plants gathered in the region. However, the levels of cesium in some wild foods (such as mushrooms) still exceed the standard values of 100 becquerels/kg in Japan [37,38]. Therefore, we should continue to communicate the risk of consuming wild foods as a focus for new residents in Kawauchi, including other areas affected by the nuclear accident.

Our study had several limitations. First, it was implemented in only one village, characterized by the advanced age of the residents and depopulation even before the FDNPP accident. Therefore, the sample size was small, which might have led to sampling bias. Second, we did not analyze the relationship between the timing of the lifting of the evacuation in their home area and the timing of their return. The evacuation order was not lifted at the same time across all of Kawauchi village. It was lifted across almost all of the central parts of the village in 2012, and lifted gradually across the remaining areas from 2012 to 2016. Temporary housing was built in the village, ready to accept residents who wanted to return from 2013 onwards. However, the timing of return might be related to the timing of the lifting of evacuation orders in their housing area. Third, we did not gather information from new residents regarding the reasons for their relocation to Kawauchi village. Among new residents, the stress-coping score and the sense of belonging in Kawauchi may differ, depending on their reasons for moving to the village, or for various other reasons. Despite the limitations of the present study, the results provide unique and valuable base data for evaluating the timing of return or new settlement and related factors in residents of an area affected by an unexpected nuclear accident, many of whom were living in the area at the time of the accident.

## 5. Conclusions

The present study identified differences in the satisfaction of current life, coping with stress, and perception of radiation risk that depended on the timing of residents’ return to or new settlement in Kawauchi following the nuclear accident. Even though they had experienced a serious nuclear accident, residents who returned to their hometown in the early phase had a high sense of satisfaction with their current life one decade after the FDNPP accident. Furthermore, they had a strong sense of belonging in Kawauchi, which is located in the area affected by the accident, and tended to think that the area had been adequately reconstructed. Interestingly, the residents who had early-phase return or settlement after the accident had a lower risk perception of genetic effects in offspring. Furthermore, we found that the number of new residents increased in all areas in Kawauchi after lifting the evacuation order. It is important to provide adequate support regarding risk communication to returned or newly settled residents, depending on the phase of reconstruction after a nuclear accident.

## Figures and Tables

**Figure 1 ijerph-19-00543-f001:**
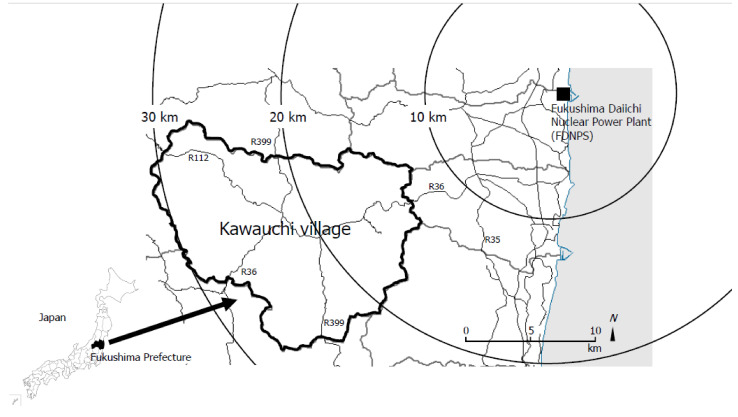
Map of Fukushima prefecture showing Kawauchi village.

**Table 1 ijerph-19-00543-t001:** Resident characteristics and survey responses in 2021.

	Overall	Group 1	Group 2	Group 3	Group 4	*p*-Value
No. of participants	374	170	84	99	21	
Sex						
Male (%)	54.5	55.3	50	59.6	42.9	0.4
Age (years)						<0.01
<50 (%)	9.8	4.1	8.3	20.1	14.3	
50s (%)	11	10	4.8	18.2	9.5	
60s (%)	32.1	40	23.8	25.3	33.4	
≥70 (%)	47.1	45.9	63.1	36.4	42.8	
Place of residence at the time of the FDNPS accident						
Kawauchi (%)	83.2	95.9	85.7	58.6	85.7	<0.01
Place of birth						
Kawauchi (%)	67.9	73.5	78.6	55.6	38.1	<0.01
Cohabitation with children						
Yes (%)	64.2	62.9	66.7	68.7	42.9	0.06
Are you satisfied with your current life?						<0.01
Satisfied (%)	23	35.3	15.5	9.1	19	
Slightly satisfied (%)	40.4	38.8	50	32.3	52.4	
Slightly unsatisfied (%)	24.6	20.6	20.2	38.4	9.5	
Unsatisfied (%)	12	5.3	14.3	20.2	19	
Do you feel that you belong to the Kawauchi village community?						<0.01
Yes (%)	35.2	45.3	39.3	20.2	9.5	
Probably (%)	38	35.9	39.3	44.4	19	
Probably not (%)	19.8	14.1	19	26.3	38.1	
No (%)	7	4.7	2.4	9.1	33	
Do you feel Kawauchi has been adequately reconstructed?						0.46
Yes (%)	9.4	10.6	8.3	8.1	9.5	
Probably (%)	33.2	36.5	35.7	25.3	33.3	
Probably not (%)	39	36.5	42.9	40.4	38.1	
No (%)	18.4	16.5	13.1	26.3	19	
Do you have concerns about consuming wild plants growing in Kawauchi?						
Yes, I have concern (%)	58.6	59.4	61.9	53.5	61.9	0.67
Do you think there may be a risk of health effects due to radiation exposure?						
Yes (%)	12.3	11.2	13.1	13.1	14.3	
Likely (%)	25.7	21.2	29.8	27.3	38.1	0.08
Unlikely (%)	40.9	40	47.6	40.4	23.8	
No (%)	21.1	27.6	9.5	19.2	23.8	
Do you believe there is a risk of genetic effects from living in Kawauchi?						<0.01
Yes (%)	18.4	15.9	16.7	23.2	23.8	
Likely (%)	27.8	21.8	34.5	32.3	28.6	
Unlikely (%)	38	40.6	45.2	28.3	33.3	
No (%)	15.8	21.8	3.6	16.2	14.3	
Post-traumatic symptoms (+)						
PCL–S score ≥12 (%)	14.2	11.8	15.5	17.2	14.3	0.65
Coping well with stress						
SOC score >59 (%)	46	53.5	48.8	30.3	47.6	<0.01

Chi-square test. Note: Time of return to Kawauchi: Group 1, <2 years; Group 2, 2–5 years; Group 3, >5 years; Group 4, not yet returned, PCL–S = post-traumatic stress disorder checklist–Specific, SOC = sense of coherence.

**Table 2 ijerph-19-00543-t002:** Regression analysis for each group in comparison with returnees >5 years.

	Group 3	Group 1	Group 2	Group 4
Time of return	>5 years	<2 years	2–5 years	Not yet returned
	OR (reference)	OR (95%CI)	OR (95%CI)	OR (95%CI)
Sex				
Male	1	0.9 (0.5–1.4)	0.7 (0.4–1.2)	0.5 (0.2–1.3)
Age in 2021				
<60 years	1	3.9 ** (2.1–7.0)	4.5 ** (2.1–9.9)	1.7 (0.6–5.2)
Place of residence at the time of the FDNPS accident				
Kawauchi	1	16.2 ** (6.9–38.0)	4.2 ** (2.0–8.7)	3.8 * (1.0–13.7)
Birthplace				
Kawauchi	1	2.2 ** (1.3–3.7)	2.9 ** (1.5–5.6)	0.6 (0.2–1.6)
Cohabitation with children				
Yes	1	1.1 (0.6–1.9)	0.9 (0.5–1.9)	3.1 * (1.1–8.5)
Satisfaction with current life				
Yes	1	4.0 ** (2.3–6.7)	2.6 ** (1.4–4.8)	3.1 * (1.1–8.7)
Sense of belonging in Kawauchi				
Yes	1	2.3 ** (1.3–4.1)	2.0 * (1.0–3.8)	0.3 ** (0.1–0.7)
Kawauchi has been adequately reconstructed				
Yes	1	1.8 ** (1.1–3.0)	1.6 (0.9–2.9)	1.2 (0.2–3.2)
Concern about consuming wild plants growing in Kawauchi				
Yes	1	1.2 (0.7–2.0)	1.4 (0.8–2.5)	1.5 (0.5–4.1)
Concern about health effects due to radiation exposure				
Yes	1	0.7 (0.4–1.2)	1.1 (0.6–2.0)	2.0 (0.8–5.5)
Risk perception of genetic effects				
Yes	1	0.4 ** (0.3–0.8)	0.9 (0.5–1.5)	1.1 (0.4–3.0)
Post-traumatic symptoms (+)				
PCL–S score ≥12	1	0.7 (0.3–1.3)	0.9 (0.4–2.0)	0.9 (0.2–3.5)
Coping well with stress				
SOC score >59	1	2.7 ** (1.6–4.5)	2.2 ** (1.2–4.0)	2.0 (0.8–5.5)

Regression analysis. Note: OR, odds ratio; 95%CI, 95% confidence interval, *p* < 0.05 *, *p* < 0.01 **.

## Data Availability

All data are available from the corresponding author on reasonable request.

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
