# Peer review of "Evaluation of Residents’ Timing of Return to or New Settlement in Kawauchi Village, at 10 Years after the Fukushima Daiichi Nuclear Power Plant Accident"

_ijerph, 2022, doi:10.3390/ijerph19010543_

Round 1

Reviewer 1 Report

Dear authors,

Your study ,,Evaluation of Residents’ Timing of Return to or Settlement in 2 Kawauchi village, at 10 Years after the Fukushima Daiichi Nu-3 clear Power Plant Accident" is a nice contribution to the disaster studies. I have some suggestions regarding your paper:

  1. The introduction part is not well written and I will suggest that part move to study area where is appropriate;
  2. You don't have a literary review part with main results of previous studies regarding your subject of research. Please, add this part.
  3. After the figure, please, put the title of your figure;
  4. Please, improve the statistical analysis part, you did not mention regression analysis, the presumption of normality, and other information important for the statistical part;
  5. Please, add sample information in the methods part;
  6. Please, try to add more information regarding the results part;
  7. Conclusions are poorly written, please, add more conclusions remarks.

Author Response

Thank you for the valuable comments and suggestions. We have revised our manuscript accordingly and provided our point-by-point responses as follows.

  1. The introduction part is not well written and I will suggest that part move to study area where is appropriate;

According to your suggestion, we have added more detail to the study area and changed evacuation orders (lines 33-50, lines 70-80). 

  1. You don't have a literary review part with main results of previous studies regarding your subject of research. Please, add this part.

According to your suggestion, we have added a review of the literature regarding the intention to return and risk perception to the Introduction (lines 50-63).

  1. After the figure, please, put the title of your figure;

We have added the title after the figure.

  1. Please, improve the statistical analysis part, you did not mention regression analysis, the presumption of normality, and other information important for the statistical part;

We have added more details of the regression analysis to the statistical analysis section of the methods (lines 147-156). We appreciate your kind advice; however, there was no need for statistics regarding the presumption of normality in this study because no comparison was made between two groups.

  1. Please, add sample information in the methods part;

We have added sample information to the Methods (lines 92-95).

  1. Please, try to add more information regarding the results part;

We have added more information to the Results (for example, lines 172-176, lines 199-202).

  1. Conclusions are poorly written, please, add more conclusions remarks.

We have added more detail to the Conclusions (lines 303-315).

Reviewer 2 Report

Please provide further detail on how this research is specifically drawn from the questionaire developed for previous studies (i.e. what were the newly added components)

The discussion references 'social' and 'psychological' consequences several times. These are broad terms and it is not always evident what, specifically, the authors mean by these terms. Please refine and/or define these in relation to literature and/or the specific measurements undertaken in the study.

Line 224/5 states that 'the levels of cesium in some wild foods (such as mushrooms) 224 still exceed the standard values'. It's unclear what is meant by 'standard values' here - the safety guidance or the average rate of cesium in these products?

Author Response

Thank you for the valuable comments and suggestions. We have revised our manuscript accordingly and provided our point-by-point responses as follows.

  1. Please provide further detail on how this research is specifically drawn from the questionnaire developed for previous studies (i.e. what were the newly added components)

We have added the new components of the questionnaire of the study to Data collection (Line 112-114).

  1. The discussion references 'social' and 'psychological' consequences several times. These are broad terms and it is not always evident what, specifically, the authors mean by these terms. Please refine and/or define these in relation to literature and/or the specific measurements undertaken in the study.

We have added the defined meaning of the ‘social’ and 'psychological' in the study to Discussion (Line 218-223).

  1. Line 224/5 states that 'the levels of cesium in some wild foods (such as mushrooms) 224 still exceed the standard values'. It's unclear what is meant by 'standard values' here - the safety guidance or the average rate of cesium in these products?

We have added a number of standard values of the cesium in foods in Japan to the session of discussion. (Line 282).

Reviewer 3 Report

First, I would like to thank the authors for conducting such an interesting study. I have a few comments that can help to improve the manuscript.

1) In the introduction section, I would like to see more info regarding the latest return policies issued by the government, if possible.

2) Also, I believe that Figure 1 is missing the caption.

3) It would be also better to include, at the end of the introduction section, the target audience of your paper, aka who will benefit from the results of your study. 

4) In the Data Collection section (2.2), it would be helpful to summarize the information in tabular format.

5) I was curious to know why the authors chose to use the chi-square test and logistic regression as methods. Did you evaluate the possibility to adopt other methods? If yes, why did you choose such methods and not others? If not, why not?

6) The first paragraph of the Discussion section should be moved in the Conclusions. 

7) The conclusion section should include suggestions for further developments.

Author Response

Thank you for the valuable comments and suggestions. We have revised our manuscript accordingly and provided our point-by-point responses as follows.

  1. In the introduction section, I would like to see more info regarding the latest return policies issued by the government, if possible.

According to your suggestion, we added more information regarding the latest return policies issued by the government (lines 33-50).

  1. Also, I believe that Figure 1 is missing the caption.

We have added the caption the figure 1.

  1. It would be also better to include, at the end of the introduction section, the target audience of your paper, aka who will benefit from the results of your study.

According to your suggestion, we added the target audience of the study to the end of the introduction section (lines 87-89).

  1.  In the Data Collection section (2.2), it would be helpful to summarize the information in tabular format.

Thank you for your useful advice, we summarized the content in Table 1 as the questionnaire information. Therefore, we didn’t change the data collection section even complicated.  

  1.  I was curious to know why the authors chose to use the chi-square test and logistic regression as methods. Did you evaluate the possibility to adopt other methods? If yes, why did you choose such methods and not others? If not, why not?

According to your suggestion, we added that why we chose to use the chi-square test and logistic regression to Statistical analyses (lines 147-156).

  1. The first paragraph of the Discussion section should be moved in the Conclusions.

According to your suggestion, we moved to the first paragraph of the Discussion section moved in the Conclusions (lines 303-305).

  1. The conclusion section should include suggestions for further developments.

According to your suggestion, we add further developments to the Conclusion (lines 313-315).

Round 2

Reviewer 1 Report

Dear authors,

Thank you for the improvements.

Kind regards

Reviewer 3 Report

Although with the minimum effort, the authors addressed my comments. The manuscript is now slightly improved. Thank you.